# Temozolomide, Simvastatin and Acetylshikonin Combination Induces Mitochondrial-Dependent Apoptosis in GBM Cells, Which Is Regulated by Autophagy

**DOI:** 10.3390/biology12020302

**Published:** 2023-02-14

**Authors:** Sima Hajiahmadi, Shahrokh Lorzadeh, Rosa Iranpour, Saeed Karima, Masoumeh Rajabibazl, Zahra Shahsavari, Saeid Ghavami

**Affiliations:** 1Department of Clinical Biochemistry, Faculty of Medicine, Shahid Beheshti University of Medical Sciences, Tehran 1985717443, Iran; 2Department of Human Anatomy and Cell Science, Max Rady College of Medicine, University of Manitoba, Winnipeg, MB R3E 0V9, Canada; 3Faculty of Medicine in Zabrze, Academia of Silesia, 41-800 Zabrze, Poland; 4Research Institute of Oncology and Hematology, Cancer Care Manitoba, University of Manitoba, Winnipeg, MB R3T 2N2, Canada

**Keywords:** statin, natural compounds, Bcl2 family proteins, intrinsic apoptosis pathway, caspase dependent apoptosis

## Abstract

**Simple Summary:**

Glioblastoma multiforme (GBM) is a deadly brain tumor. The current chemotherapy strategies (including using temozolomide (TMZ)) is not very effective for GBM patients. Therefore, finding a new therapeutic strategy is demanding in the field of GBM. In our current investigations, we used an innovative combination of TMZ, and an FDA-approved, cholesterol-lowering medication (simvastatin), and a Chinese herbal medicine derivative (acetylshikonin) in GBM cell lines. Our investigation showed that the triple-combination treatment (TMZ/Simva/ASH) induced significantly more cell death via damaging the energy engine of the cells (mitochondria). In addition, inhibition of the cellular self-eating mechanism (autophagy) sensitizes the GBM to triple-combination-induced cell death. Overall, the current research may open new avenues in the treatment of GBM patients in the long term.

**Abstract:**

Glioblastoma multiforme (GBM) is one of the deadliest cancers. Temozolomide (TMZ) is the most common chemotherapy used for GBM patients. Recently, combination chemotherapy strategies have had more effective antitumor effects and focus on slowing down the development of chemotherapy resistance. A combination of TMZ and cholesterol-lowering medications (statins) is currently under investigation in in vivo and clinical trials. In our current investigation, we have used a triple-combination therapy of TMZ, Simvastatin (Simva), and acetylshikonin, and investigated its apoptotic mechanism in GBM cell lines (U87 and U251). We used viability, apoptosis, reactive oxygen species, mitochondrial membrane potential (MMP), caspase-3/-7, acridine orange (AO) and immunoblotting autophagy assays. Our results showed that a TMZ/Simva/ASH combination therapy induced significantly more apoptosis compared to TMZ, Simva, ASH, and TMZ/Simva treatments in GBM cells. Apoptosis via TMZ/Simva/ASH treatment induced mitochondrial damage (increase of ROS, decrease of MMP) and caspase-3/7 activation in both GBM cell lines. Compared to all single treatments and the TMZ/Simva treatment, TMZ/Simva/ASH significantly increased positive acidic vacuole organelles. We further confirmed that the increase of AVOs during the TMZ/Simva/ASH treatment was due to the partial inhibition of autophagy flux (accumulation of LC3β-II and a decrease in p62 degradation) in GBM cells. Our investigation also showed that TMZ/Simva/ASH-induced cell death was depended on autophagy flux, as further inhibition of autophagy flux increased TMZ/Simva/ASH-induced cell death in GBM cells. Finally, our results showed that TMZ/Simva/ASH treatment potentially depends on an increase of Bax expression in GBM cells. Our current investigation might open new avenues for a more effective treatment of GBM, but further investigations are required for a better identification of the mechanisms.

## 1. Introduction

Among the known adult malignant brain tumors, glioblastoma multiforme (GBM) is the most aggressive glioma [1]. Despite advances in medical technologies, the average life-span of GBM patients after surgery and chemotherapy is 14–15 months [2]. Although the annual incidence rate of GBM is about 3.19 to 4.17 cases per 100,000 people [3], its high mortality rate and poor prognosis encourage researchers to find more effective treatments [4].

TMZ is widely used as the first line of chemotherapy for GBM patients [5,6]. It is an alkylating agent [7,8] and induces DNA damage, cell cycle arrest and apoptosis activation [8,9,10]. TMZ also induces autophagy, which could have cytoprotective effects on the target cells [9,10,11].

The cell fate is determined via different mechanisms, including apoptosis, autophagy, and the unfolded protein response (UPR) [12,13]. These pathways are initiated and characterized by multiple stimuli and signaling mediators [14,15]. The main function of autophagy relates to the cell’s protection and survival, while under some situations it leads to cell death [16,17]. There are different molecular pathways for crosstalk between the apoptosis and autophagy pathways [18,19,20]. Autophagy can be a pro- or anti-apoptotic mechanism based on the type of cell model, organism and the stimuli [21,22,23,24].

Statins are cholesterol-lowering drugs, with several pleiotropic effects, including their impact on cancer [25]. In recent years, both basic and clinical scientists have focused their investigations on the potential role of statins in cancer therapy. Recent studies have suggested that statins could have beneficial effects and impacts on the response to chemotherapy in cancer patients [9,26,27,28,29]. A recent investigation has showed that simvastatin combined with TMZ increases the survival of GBM patients [30]. Our team has recently discovered that simvastatin induces apoptotic cell death in glioblastoma, non-small cell lung carcinoma, breast cancer, and neuroblastoma cell lines via a depletion of geranylgernaly pyrophosphate [31,32]. We have later showed that simvastatin sensitizes GBM tumor cell lines (U87 and U251) and primary patient-derived GBM cells to TMZ-induced apoptosis, via an inhibition of autophagy flux [10] and UPR [20]. 

Recently, Shikonin [32], a highly lipophilic compound from the Lithospermum erythrorhizon root that is commonly used in Chinese folklore remedies for its anti-inflammatory and pleiotropic effects, has been introduced as an antitumor agent [33,34,35,36,37]. Besides the strong anticancer feature of Shikonin and its analogs, it also has the ability of circumventing cancer drug resistance [38,39]. Shikonin derivatives, such as acetylshikonin [10], have shown cytotoxic and anticancer effects [40]. A few recent investigations also showed the impact of SHK (alone or in combination with TMZ) in targeting GBM tumor cells by inducing significant apoptotic cell death and decreasing their proliferation [34,41].

In the current investigation, we expand our previous investigations and used triple treatments of ASH, TMZ and Simvastatin (Simva) to target GBM tumor cells (U251 and U87). We focused our investigation on the mechanism of combination therapy on GBM cells via a cross talk of apoptosis and autophagy. 

## 2. Materials and Methods

### 2.1. Reagents and Drugs

Acetylshikonin (CAS Number: 24502-78-1) and simvastatin (CAS Number: 79902-63-9) were purchased from Chem Face China Company. Temozolomide (CAS Number: 85622-93-1), propidium iodide (CAS Number: 25535-16-4), 3-[4,5-dimethylthiazol-2-yl]-2,5 diphenyl tetrazolium bromide (MTT) (CAS Number: 298-93-1), acridine orange (CAS Number: 65-61-2), bafilomycin A1 (Cat#B1793-10UG) and anti-rabbit LC3β antibody (Cat#L7543—100UL) were purchased from Sigma-Aldrich Co. Anti-rabbit p62 antibody (Cat#39749), anti-rabbit Beclin-1 antibody (Cat#3738), anti-mouse Bcl2 antibody (Cat#15071), anti-rabbit Mcl-1 antibody (Cat#4572), anti-rabbit Bcl-XL antibody (Cat#2762), and anti-rabbit GAPDH antibody (Cat#2118) were purchased from Cell Signaling Company. The secondary antibodies, anti-rabbit HRP-conjugate and anti-mouse HRP-conjugate, were purchased from Sigma-Aldrich (Oakville, ON, Canada) as well. The enhanced chemiluminescence (ECL) (CAS Number: 12630) was acquired from Cell Signaling Technology Co. (Beverly, MA, USA). The bicinchoninic acid (BCA) protein assay kit was obtained from Thermo Fisher Scientific (Winnipeg, MB, Canada).

### 2.2. Cell Lines, Culture, and Treatment

U87 and U251 human GBM cells were obtained from Bon yakhteh Company (Bon yakhteh, Tehran, Iran) and cultured in Dulbecco’s Modified Eagle Medium (DMEM) (Bio Idea, Tehran, Iran, Cat #: DB9696) (low-glucose, high glutamine for U87 and high-glucose for U251), supplemented with 10% fetal bovine serum (FBS) (Bio Idea, Tehran, Iran, Cat #: BI-1201) and 1% penicillin–streptomycin (Bio Idea, Tehran, Iran, Cat #: BI-1203). Cells were maintained in a 5% CO_2_ incubator with 95% humidity at 37 °C.

### 2.3. Cytotoxicity Assay

To evaluate the cytotoxicity effects of temozolomide, ASH and Simva and their combination on U87 and U251 GBM cell lines, MTT viability tests were done based on our established method [8,10,31]. Four different time points (24, 48, 72 and 96 h) and a range of concentrations (TMZ 50–250 µM, ASH 0.5–25 µM, Simva 0.5–20 µM) were tested. After finding the best time point and the IC50 concentration of each drug, the combination therapy of TMZ/Simva and TMZ/Simva/ASH on GBM cells was evaluated. For combination therapy, Simva was pretreated for 4 h. U87 and U251 cells were both briefly cultured in 96-well plates (6000 cells per well) and treated with different concentrations of the drugs. After the treatment time point, 20 µL of MTT dye, 3-[4,5-dimethylthiazol-2-yl]-2,5 diphenyl tetrazolium bromide was added to each well and incubated in the 37 °C incubator for 4 h. The medium was then aspirated and 200 µL of DMSO was added to each well and incubated at room temperature for 15 min in the dark. Finally, the optical absorbance was read at 570 nm by multiplate reader (Synergy, Biotek, Winooski, VT, USA). 

### 2.4. Apoptosis Assay Using Flow Cytometry

The cell death mechanism evaluation was measured using the Nicolleti method [42,43,44]. Both cell lines were briefly cultured in 6-well plates (100,000 cells per well) and pre-treated with Simva 1 µM and 2.5 µM for U251 and U87, respectively, for 4 h. These were the least toxic concentrations of Simva for each cell line. Then, TMZ 100 µM and ASH 1.5 µM were added. After 72 h, the cells were detached by an EDTA buffer to minimize cell damage, and then washed with PBS. Afterward, the PI lysis buffer, including 0.1% Triton X-100, 40 µg/mL propidium iodide, 1% sodium citrate and 0.5 mg/mL RNase A, was added to the obtained pellets and incubated at 37 °C for 35 min. Finally, sub G1 population area indicated the apoptotic nuclei levels by flow cytometry (FACScalibur flow cytometer, BD Biosciences, USA). All the results were acquired in 10,000 event count.

### 2.5. Immunoblotting

We did Western blotting according to our previous investigations [45,46,47]. After treatment with the different compounds (TMZ/Simva-ASH, ASH, Bafilomycin A1 (Baf-A1)) at the indicated time point (72 h), the cells were gently washed with PBS, and the pellets were collected. The pellets were suspended in the lysis buffer, including NaCl 150 Mm, Triton 1%, Tris 50 Mm, protease inhibitor and a phosphatase inhibitor cocktail (Cat#P5726-Sigma), and then sonicated 5 times for 3 s. Equal amounts of proteins were loaded onto the electrophoresis SDS-PAGE gel. The proteins were then transferred to PVDF membranes (Sigma; # IPVH00010). The blocking step was performed in 5% fat-free milk overnight. The primary antibodies (Bax, Bcl-2, LC3β-II and p62) were also diluted, based on the manufacture’s protocol to add the membranes and incubate at 4 °C overnight. The next step was the incubation of the membranes with the corresponding secondary antibodies at room temperature for 90 min. Finally, the membranes were exposed to enhanced chemiluminescence (ECL) reagents and developed by the ChemiDocTM MP imaging system (Bio-Rad, Hercules, CA, USA). To quantify the intensity of the bands, the Image Lab densitometry software was utilized and normalized by GAPDH protein values to correct the possible errors during protein loading.

### 2.6. Caspase3/7 Activation Assay

To assess the activity and catalytic function of caspase3/7 (DEVD-ase), a Cayman fluorescence assay kit was used based on our adopted established protocol [10,31,48]. Briefly, the cells were seeded in 96-well plates to reach about 50% confluence, to prevent high non-specific fluorescence, and were then treated with TMZ (100 µM), Simva (1 and 2.5 µM for U251 and U87, respectively), ASH (1.5 µM), TMZ/Simva and TMZ/Simva/ASH, for 48 h based on our experimental protocol (OEP). For the combinations, Simva was pretreated for 4 h. All reagents were freshly prepared, including the assay buffer, caspase3/7 substrate, active caspase-3 positive control and caspase3/7 inhibitor solution. The plates were centrifuged at 800× *g* for 5 min, followed by aspiration of the supernatant, addition of the assay buffer and another centrifugation. Later, lysis buffer was added, followed by orbital shaking and centrifuging (800× *g*). Caspase3/7 inhibitor, active caspase-3 positive control and caspase3/7 substrate solutions were then added (30 min, room temperature, and a dark place). The fluorescence intensity of each well was measured by BioTek Cytation 3 cell imaging multi-mode microplate reader (Biotek Cytation 3, USA) (excitation wavelength: 485 nm, emission wavelength: 535 nm).

### 2.7. Reactive Oxygen Assay

ROS generation was detected by a Cayman ROS detection cell-based Dihydroethidium (Hydroethidine, DHE) assay. The cells were cultured in the black tissue culture-treated 96-well plates, where their confluence reached about 70% after 24 h to avoid overgrown cells. The treatment was done at two different time points (38, 60 h). The cell-based assay buffer and DHE, N-Acetyl cysteine and antimycin assay reagents were prepared based on the kit protocol. All wells were aspirated. The assay buffer and ROS-staining buffer were then added, respectively. N-Acetyl cysteine and antimycin were used as the negative and positive controls, respectively. The kit’s instruction was followed for the incubation times, fluorescence was measured (excitation wavelength of 480–520 nm and emission wavelength of 570–620 nm), and microscopic images were taken by BioTek Cytation 3 cell imaging multi-mode microplate reader (Biotek Cytation 3, USA) and analyzed based on our established protocols [10].

### 2.8. Determination of Mitochondrial Membrane Potential (MMP) 

Loss of mitochondrial membrane potential is considered a substantial parameter of cell function. Thus, the measurement of changes in the MMP is very important to follow the apoptotic process. After seeding, a Cayman JC-1 mitochondrial membrane potential assay kit was used on the cells in the 96-well black culture plates, treated for 48 h. The cells’ confluence was 80% at the time of staining. JC-1 was added to each well and the assay buffer was used after incubation. The test was performed based on the kit directions. The healthy cells with JC-1 J-aggregates were detected with a Rhodamine filter (excitation/emission: 540/570), while the apoptotic cells, with mainly JC-1 monomers, were detectable with an FITC filter (excitation/emission 485/540) by BioTek Cytation 3 cell imaging multi-mode microplate reader (Biotek Cytation 3, USA).

### 2.9. Acridine Orange Acidic Vacuole Assay

To detect the acidic vesicular organelles [49], which are essential markers of late autophagy [50], an acridine orange (AO) staining assay was performed. AO induces green fluorescence in cytosolic and nuclear parts of the cells, but upon fusion into the acidic environment, such as lysosomes, it becomes protonated and emits an intense red fluorescence. Therefore, the ratio of red to green fluorescence intensity can be a suitable marker for AVO formation. Cells were seeded in 96-well plates and treated for 72 h. They were then stained with AO (final concentration of 1 μg/mL) and incubated at 37 °C for 10 min (dark and room temperature). After washing with PBS twice, the microscopic images were taken, and the green and red fluorescence was read at 550 and 650 nm of emission wavelengths by BioTek Cytation 3 cell imaging multi-mode microplate reader (Biotek Cytation 3, USA), respectively. 

### 2.10. Statistical Analysis 

The results were represented as the means ± SD and statistical differences were analyzed by a one-way or two-way ANOVA, using Graph Pad Prism 8. A *p*-value < 0.05 implies the significance values. All the experiments were done in at least three biological replicates.

## 3. Results

### 3.1. TMZ/Simva/ASH Combination Treatment Induces More Cell Death Compared to Single Treatment in Human GBM Cells

Our previous investigations have showed that the co-treatment of Simva/TMZ increased cytotoxicity effects in comparison to TMZ and Simva single treatment in GBM cells (U251 and U87) [10,20]. ASH was used in combination with Simva/TMZ in our current investigation, as it induces cell death in different tumor models, including GBM cells, [51,52,53] and crosses the blood-brain barrier [53]. We investigated if this combination induces more cell death compared to the single therapies and the Simva/TMZ combination therapy.

Initially we determined the cytotoxic effects of ASH, TMZ, and Simva in U251 and U87 cells. We treated these cells with different concentrations of ASH, TMZ, and Simva (ASH (0.5–25 µM), TMZ (50–250 µM), and Simva (0.5–20 µM)) at different time points (24, 48, 72 and 96 h). Our results showed that all concentrations of Simva induced significant cell death (*p* < 0.0001), except 0.5 µM at 24 and 48 h (*p* > 0.05) in U251 (Figure 1A–D) and U87 (Appendix A) cells. We also determined that all concentrations of TMZ induced significant cell death (*p* < 0.0001, *p* < 0.05) in U251 (Figure 1E–H) and U87 (Appendix A) cells, except for 50 µM at 24 and 48 h (*p* > 0.05). Our investigations also showed that all concentrations of ASH induced significant cell death (*p* < 0.0001, *p* < 0.05, *p* < 0.01) in U251 (Figure 1I–L) and U87 (Appendix A) cells. The 0.5 µM concentration of ASH did not induce significant cell death at 24 h in U87 cells.

In the next step we identified the cytotoxic effects of Simva/TMZ/ASH in GBM cells. Based on our previous investigations [10,20] and the results of single treatments, we used Simva (1 µM, U251; and 2.5 µM, U87), TMZ (100 µM) and ASH (1.5 µM) for 72 h based on the minimum toxic dose (<25%). Our findings showed that the TMZ/Simva/ASH combination therapy induced significantly higher cell death compared to single therapies (TMZ, Simva, ASH) in both U251 (Figure 2A–D) and U87 (Appendix A) at 72 h (*p* < 0.0001). Furthermore, the results showed that ASH significantly increased the toxicity effects of TMZ/Simva in both U251 and U87 cells at 72 h (*p* < 0.0001), (Figure 2A–D, Appendix A).

### 3.2. TMZ, Simva, ASH, TMZ/Simva and TMZ/Simva/ASH Treatments Induce Caspase-Dependent Apoptosis in GBM Cells

In our study, we used TMZ, Simva, ASH, TMZ/Simva and TMZ/Simva/ASH treatments to induce apoptosis in GBM cells. Concentrations of 1 µM Simva for U251, 2.5 µM Simva for U87, 100 µM TMZ (U251 and U87), and 1.5 µM ASH (U251 and U87) were used. The results showed that the apoptotic cell population for TMZ/Simva was 21 and 24% in U87 and U251, respectively, which is significantly higher than for TMZ (12%) and Simva (6%) single treatments in both cells at 72 h (*p* < 0.0001) (Figure 2E). TMZ/Simva/ASH combination significantly increased apoptosis compared to TMZ/Simva and ASH single therapy in U87 and U251 cells (*p* < 0.0001), (Figure 2F,G).

Caspase-3/-7 are the final executing caspases and induce the last steps in nuclear events related to apoptosis [37,54,55]. We measured their activity at 60 h to investigate the impact of caspase activation in TMZ, Simva, ASH, TMZ/Simva, and TMZ/Simva/ASH-induced apoptosis in GBM cells. Our results showed that caspase-3/-7 were significantly activated in all treatment conditions in U87 (Figure 2H) and U251 (Figure 2I) cells (*p* < 0.0001). Our results also showed that TMZ/Simva and TMZ/Simva/ASH combination therapy showed significantly higher caspase-3/-7 activation in both cell lines compared to single treatment (*p* < 0.0001) (Figure 2H,I). Additionally, the TMZ/Simva/ASH combination treatment induced significantly higher activation of caspase-3/-7 compared to TMZ/Simva treatment in both U87 and U251 cells (*p* < 0.0001) (Figure 2H,I). These findings correlated with the higher apoptosis induction in triple-combination treatments for both cell lines compared to TMZ/Simva treatment (*p* < 0.0001) (Figure 2H,I).

### 3.3. TMZ, Simva, ASH, TMZ/Simva and TMZ/Simva/ASH Treatments Increase Reactive Oxygen Species and Decrease Mitochondrial Membrane Potential in GBM Cells

The presence of ROS and a decrease in mitochondrial membrane potential are hallmarks of apoptosis [24,49,56,57,58,59]. Therefore, we have measured the amount of ROS and the mitochondrial membrane potential in different experimental conditions.

We assessed ROS in TMZ, Simva, ASH, TMZ/Simva and TMZ/Simva/ASH treatments at two different time points (38 h and 60 h). We observed a significant increase in ROS in all treatments, in both cell lines and time points in U251 (Figure 3A–D) and U87 (Appendix A) compared to the time-match vehicle control (*p* < 0.0001). However, ROS did not significantly increase in the Simva treatment in U87 cells at 60 h (Appendix A). Both TMZ/Simva and TMZ/Simva/ASH combination treatments significantly increased ROS (*p* < 0.0001) compared to single treatment, except for TMZ/Simva compared to TMZ treatments in U87 at 38 h (Appendix A) (*p* > 0.05).

Our investigations also showed that all treatments (TMZ, Simva, ASH, TMZ/Simva and TMZ/Simva/ASH) significantly reduced mitochondrial membrane potential in both U251 (Figure 3E,F) and U87 (Appendix A) (*p* < 0.0001). Both combination treatments (TMZ/Simva, TIMZ/Simva/ASH) significantly reduced mitochondrial membrane potential in U251 (Figure 3E,F) and U87 (Appendix A) compared to single treatment (*p* < 0.0001).

### 3.4. The Impact of TMZ, Simva, ASH, TMZ/Simva and TMZ/Simva/ASH Treatments on Autophagy in GBM Cells

Autophagy and apoptosis are interconnected, as autophagy can positively or negatively control the apoptosis induction, based on the type of stimuli and the cell models [22,57,60,61]. In our recent investigations we have shown that Simva inhibits the fusion of autophagosomes and lysosomes, and increases the apoptotic response to TMZ treatment in GBM cell lines and primary patient-derived GBM tumor cells [10]. Therefore, in our current investigation, we addressed the impact of autophagy in response to TMZ/Simva/ASH combination treatment. We first evaluated AVOs using AO staining under different experimental conditions, including TMZ, Simva, ASH, TMZ/Simva and TMZ/Simva/ASH at 72 h. The results showed that the number of AVOs significantly increased in all experimental conditions, including TMZ and ASH treatments (statistical significance of *p* < 0.05) and Simva, TMZ/Simva, and TMZ/Simva/ASH treatments (statistical significance of *p* < 0.001) in both U87 and U251 cells (Figure 4A–C). The significant increase of AVO-positive cells in the combination treatments, including Simva and ASH, guided us to investigate the potential mechanism that led to this result. In our previous investigation, our team has shown that Simva co-treatment with TMZ increased the number of autophagosomes and AVOs, because of the inhibition of autophagosome and lysosome fusion in GBM cells [10]. Our current results showed that a combination of TMZ/Simva/ASH significantly increased the number of AVOs compared to TMZ/Simva treatment (Figure 4A–C).

In the next step, we evaluated the biogenesis of autophagosome and autophagy flux in different experimental conditions (ASH, TMZ/Simva/ASH) using Baf-A1 and inhibition of autophagy flux (5 nM, 72 h). The results showed that the ASH treatment did not significantly change LC3β-II/LC3β-I and p62 in both U87 and U251 cells (*p* > 0.05) (Figure 4D,G,H,L,M). In addition, our results showed that there is no significant difference in LC3β-II/LC3β-I and p62 between Baf-A1 treatment and Baf-A1/ASH (*p* > 0.05) (Figure 4D,G,H,L,M). TMZ/Simva/ASH non-significantly (*p* > 0.05) increased LC3β-II/LC3β-I and p62, while Baf-A1 co-treatment significantly changed LC3β-II/LC3β-I and p62 in both U87 and U251 cells (*p* < 0.01, *p* < 0.001) (Figure 4D,G,H,L,M). Therefore, ASH treatment increases the number of AVO-positive cells via the fast turnover of autophagosomes, while TMZ/Simva/ASH increases AVO-positive cells via the partial inhibition of autophagy flux in GBM cells (U87 and U251).

### 3.5. Autophagy Flux Inhibition Increases Simva, TMZ, ASH and Combination Triple Treatment-Induced Cell Death in GBM Cells

We have shown that the TMZ/Simva/ASH combination treatment induced a partial block of the autophagy flux in GBM cells (Figure 4). We further investigated the impact of autophagy flux inhibition in TMZ, Simva, ASH, and TMZ/Simva/ASH-induced cell death in GBM cells. We used chloroquine (CQ) as the autophagy flux inhibitor [61,62]. We first investigated the cytotoxic effects of CQ (25–200 µM) at different time points (24, 48 and 72 h) using an MTT assay to optimize the lowest concentration that inhibits the flux and has the lowest cytotoxic effect in GBM cells. The results are summarized in Figure 5A–C (U251) and Appendix A (U87), which show that the least toxic concentration of CQ that still inhibits flux (data was not showed) are 25 µM and 50 µM in U87 and U251 cells, respectively. Therefore, we used a combination of CQ (25 µM for U87, 50 µM for U251) with TMZ (100 µM), Simva (2.5 and 1 µM for U87 and U251, respectively), ASH (1.5 µM) and TMZ/Simva/ASH. CQ (autophagy flux inhibition) in combination with Simva (Figure 5D–F, Appendix A), TMZ (Figure 5G–I, Appendix A), ASH (Figure 5J–L, Appendix A) and TMZ/Simva/ASH (Figure 5M–O, Appendix A) significantly (*p* < 0.0001) decreased the cell viability in both U251 and U87 cell lines. Overall, it can be concluded that autophagy flux inhibition increased the triple-combination-treatment-induced cell death in GBM cells.

### 3.6. The Role of Bcl2 Pro- and Anti-Apoptotic Family Proteins in TMZ/Simva/ASH-Induced Cell Death in GBM Cells

Bcl2 family proteins play an important role in the regulation of apoptotic cell death related to mitochondria [58,63,64]. Our results showed that TMZ, Simva, ASH, TMZ/Simva, and TMZ/Simva/ASH treatments increased ROS and decreased the mitochondrial membrane potential in U251 (Figure 3A–F) and U87 (Appendix A) GBM cells. Our previous investigations showed that Bcl2 family proteins are involved in the regulation of statins and TMZ-induced cell death in GBM cell [10,31]. Our current investigation showed that Baf/TMZ/Simva/ASH combination significantly increased the pro-apoptotic protein Bax in U87 cells (Figure 6A,B) (*p* < 0.001), while the treatment non-significantly increased Bax expression in U251 (Figure 6A,F) (*p* > 0.05). On the other hand, TMZ/Simva/ASH did not significantly change the expression of anti-apoptotic proteins (Bcl2, Bcl-XL, Mcl-1) in U87 (Figure 6A,C–E) and U251 (Figure 6A,G–I) cells (*p* > 0.05). Therefore, it can be concluded that TMZ/Simva/ASH-induced cell death in GBM cells is potentially dependent on pro-apoptotic Bax protein expression.

## 4. Discussion

Many chemotherapy approaches are not effective towards reaching the desired outcomes in GBM patient treatment [65,66]. Our team has recently investigated the efficiency and mechanisms of combination Simva and TMZ treatments in GBM, and showed that Simva sensitizes GBM cells to TMZ-induced apoptosis [10,20,22]. TMZ resistance, as a major problem in GBM patient therapeutic strategies, has been investigated for years. Moreover, several compounds and drugs were utilized to enhance the efficacy of TMZ, while attenuating the resistance to TMZ, to achieve the best approach to treating GBM patients [65,66,67]. Recent investigations have showed that taking statins for a long period of time can potentially increase the survival of the cancer patients (including GBM) and improve their response to different chemotherapy medications [9,27,30,68]. A significant synergistic effect of statins has been reported in combination with chemotherapy compounds, including cisplatin, 5-Fluorouracil (5-FU), doxorubicin and paclitaxel [69]. A recent investigation has also showed that lovastatin sensitizes GBM cells (U251 and U87) to TMZ-induced apoptosis via an inhibition of autophagy flux [70]. These results support our recent findings about the mechanisms of sensitization of GBM cells to TMZ-induced apoptosis using Simva [10]. Several other investigations have showed that simvastatin induces apoptosis in breast cancer, chronic myeloid leukemia (CML) and lung cancer cells via changing the balance between pro- and anti-apoptotic Bcl2 family proteins [69]. Co-treatment of Simva and flavons decreases chemo-resistance to doxorubicine via the degradation of multidrug resistance (MDR) proteins in human colon cancer cells [71]. In another investigation, it has been shown that liposomal Simva sensitizes C26 colon carcinoma cells to liposomal 5-FU via an inhibition of tumor angiogenesis in vivo [72]. Simvastatin also sensitizes A549 non-small cell lung cancer to Sulindac- or Pemetrexed- (multi-target antifolate medication) induced caspase-dependent apoptosis via damaging mitochondria and increasing ROS [73,74].

Our previous studies revealed that simvastatin triggers the intrinsic apoptosis in various human cancer cells, including GBM, via an inhibition of the mevalonate cascade, with subsequent targeting of geranylgerany pyrophosphate prenylation precursors [31]. We later showed that the TMZ/Simva combination treatment increased apoptosis compared to the TMZ and Simva single treatments in GBM cells. Our investigations showed that Simva sensitizes GBM cells and GBM patient-derived tumor cells to TMZ-induced apoptosis via an inhibition of autophagy flux and UPR induction [10,20]. Our previous investigations have showed that Simva/TMZ combination inhibits the fusion of autophagosomes and lysosomes in GBM cells and sensitizes GBM cells to TMZ-induced apoptosis. In addition, Simva/TMZ combination activates PERK and IRE1 arm of UPR, and further affects the autophagy flux via the UPR pathway in GBM cells [10,20].

In the current investigation, for the first time, we have assessed the mechanisms of ASH-induced apoptosis and later provided mechanisms which are involved in TMZ/Simva/ASH-induced apoptosis in GBM cells. In summary, our results showed that ASH and its triple-combination treatment (TMZ/Simva/ASH) increased cellular ROS and decreased the mitochondrial membrane potential with subsequent caspase-dependent apoptosis induction in GBM cells. In addition, our results showed that ASH induced autophagy, while TMZ/Simva/ASH partially inhibited autophagy flux in GBM cells. Furthermore, blocking the autophagy flux increased ASH and TMZ/Simva/ASH-induced cell death in GBM cells, which clearly shows that both ASH and TMZ/Simva/ASH-induced cell death are dependent on the autophagy pathway.

ASH induces apoptosis in a wide range of tumor cells, such as ROS-dependent apoptosis in oral squamous-cell carcinoma cells (Ca9-22) [7]. Recent investigations have shown that ASH increases ROS and nuclear damage with a subsequent nuclear translocation of FOXO3 and induced caspase-dependent apoptosis in osteosarcoma U2OS, renal-cell carcinoma and colorectal cancer HCT-15, and LoVo cells [75,76,77]. It has also been reported that ASH induces dose-dependent apoptosis via an activation of caspase-3/-7 and -9 in chondrosarcoma cell lines [78]. They also showed that MAPK activation is involved in ASH-induced apoptosis in colorectal cancer cells [78]. On the other hand, ASH induces caspase-dependent apoptosis via ROS and an inhibition of NF-κB in K562 leukemia cells [79]. The combination of Erlotinib and Shikonin (and its derivatives) has been recently assessed in GBM cells. The cytotoxicity results showed a synergic effect of Shikonin and its derivatives, including ASH with Erlotinib in GBM cells (U87, BS153, A431 and DK-MG) [80]. In conclusion, the ASH-induced ROS hampers tumor cell proliferation, leading to the caspase-dependent apoptosis in cancer cells, including GBM cells. In addition, combination therapy with Shikonin, or its derivatives, improves the response of cancer cells to chemotherapy agents and induces a higher apoptotic cell death compared to single chemotherapy strategies.

Previous investigations have shown that an increase of pro-apoptotic Bcl2 family proteins or a decrease in anti-apoptotic Bcl2 protein expression might be involved in mitochondrial damage, increase in cellular ROS and decrease in mitochondrial membrane potential [31,44,48,63,81,82,83]. It has been shown that ASH induces apoptosis in osteosarcoma U2OS cells, A498 and ACHN (human RCC cell lines), colorectal cancer HCT-15 and LoVo cells, and human leukemia cell line K562, via changing the Bcl2 family proteins [75,76,77,79]. Interestingly, our investigations showed that ASH did not significantly change Bax, Bcl2, Bcl-XL and Mcl-1 expression, while it decreased the mitochondrial membrane potential, increased ROS, activated caspase-3/-7 and induced apoptosis in GBM cell lines (U87 and U251). In addition, our investigation has revealed, for the first time, that TMZ/Simva/ASH induces mitochondria-dependent apoptosis without significant changes in the expression of anti-apoptotic Bcl2 family proteins (Bcl2, Bcl-XL, and Mcl-1), while non-significantly increasing Bax pro-apoptotic protein in U87 and U251 cells. Therefore, our findings indicate that ASH and TMZ/Simva/ASH mitochondria-induced apoptosis in GBM cells might be dependent on other Bcl2 family proteins or could be triggered by other mitochondrial factors, such as Smac/Diablo and Omi/HtrA2 [63]. Interestingly, bafilomycin-A1 increases anti-apoptotic protein Mcl-1 in both U87 and U251 cells and Bcl-XL expression in U251 cells (which could not be mechanistically explained and needs further investigations to justify its mechanisms).

In our recent investigations, our team has showed that autophagy is a regulator of apoptosis induction in different types of cells, including airway mesenchymal cells [44,49,64], atrial fibroblasts [84], primary cardiac myofibroblasts [49], human alveolar rhabdomyosarcoma cells [8], HCT116, colorectal cancer cell line [42,43] and GBM cells [10]. Recent investigations have also showed that ASH induces autophagy via the PI3/AKT pathway, which controls its apoptosis induction in acute myeloid leukaemia (AML) cells [85]. Our current investigations also showed that ASH induces simultaneous autophagy and apoptosis, and autophagy flux inhibition increases ASH-induced cell death in GBM cells. For the first time, we have also showed that TMZ/Simva/ASH partially inhibits the autophagy flux in GBM cells, while further induction of the autophagy flux inhibition significantly increased triple-combination-therapy-induced cell death in these cells. Overall, our investigations showed that the autophagy flux plays an important role in both ASH and TMZ/Simva/ASH-induced cell death in GBM cell lines.

In our future investigations, we will try to address the impact of UPR in TMZ/Simva/ASH-induced apoptosis, as it has been recently showed that UPR is involved in ASH-induced apoptosis [86]. As TMZ, Simva, and ASH are not selective to cancer cells and will have side effects, we will use the backbone of our recent Simva-loaded nanoparticle, which specifically binds to GBM cells for potential delivery of the combination therapy to GBM tumor cells, to decrease adverse effects of these compounds in normal cells [87]. We will attempt to load both Simva and ASH on these nanoparticles to increase the efficiency of targeting GBM with a triple-combination therapy. We will also move towards both flank and xenograft GBM animal models to test different combination therapies and investigate the impact of these approaches on animal models, and be closer to clinical applications of these medications.

## 5. Conclusions

In this study, we assessed the efficacy of TMZ/Simva/ASH combination on GBM cancer cells compared to single therapies. The changes in the cell viability, sub-G1 cell population, ROS levels, caspase activation and mitochondrial membrane potential were explored to demonstrate the apoptosis cell death. AVOs, LC3II and p62 markers were also examined as the main autophagy markers. According to the findings, TMZ and ASH induced both mitochondrial apoptosis and autophagy. The triple-combination treatment induced apoptosis and partially inhibits autophagy flux. Moreover, the triple-treatment-induced apoptosis was significantly higher than the single therapies. Our results also showed that TMZ, Simva, ASH and TMZ/Simva/ASH-induced apoptosis increases via the autophagy flux inhibition. TMZ/Simva/ASH combination treatment induces apoptotic cell death in shorter time points compared to TMZ treatment.

## Figures and Tables

**Figure 1 biology-12-00302-f001:**
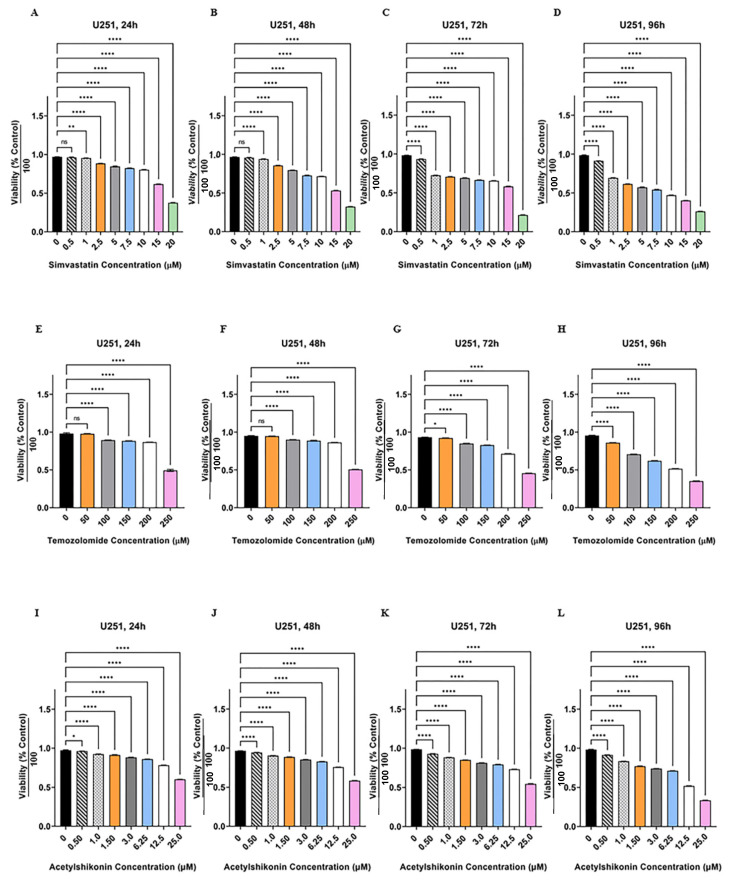
Simvastatin, temozolomide and acetylshikonin induce cell death in human glioblastoma cells. U251 cells were treated with different concentrations of Simva (0.5–20 µM, **A**–**D**), TMZ (50–100 µM, **E**–**H**), and ASH (0.5–25 µM, **I**–**L**), at four time points (24, 48, 72 and 96 h). The cytotoxicity of each treatment was measured using an MTT assay. The cell viability was measured (ns = non-significant; * = *p* < 0.05; ** = *p* < 0.01; **** = *p* < 0.0001) and compared to the time-match vehicle control. Simva induced a significant decrease in cell viability (*p* < 0.0001) for all concentrations at all time points, except 0.5 µM at 24 and 48 h (*p* > 0.05) and 1 µM at 24 h (*p* < 0.01). TMZ induced significant cell death (*p* < 0.0001) for all concentrations at all time points, except 50 µM at 24 and 48 h (*p* > 0.05) and 50 µM at 72 h (*p* < 0.05). ASH induced significant cell death (*p* < 0.0001) for all concentrations at all time points, except 0.5 µM at 24 h (*p* < 0.05). The represented findings are from nine replicates in three independent biological assays and are showed as the mean ± SD.

**Figure 2 biology-12-00302-f002:**
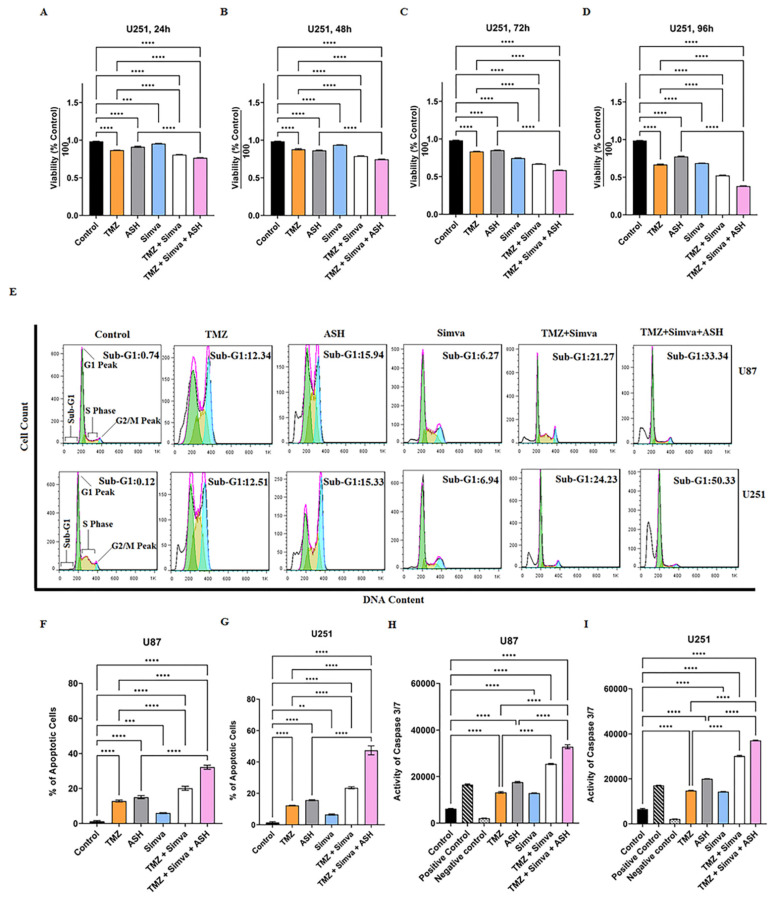
Simvastatin, temozolomide, acetylshikonin and their combination treatments induce significant apoptotic cell death in human glioblastoma cells. U251 cells were treated with TMZ 100 µM, Simva 1 µM, ASH 1.5 µM, TMZ/Simva and TMZ/Simva/ASH at 24, 48, 72 and 96 h time points. Cell viability was measured using an MTT assay and compared to the time-match vehicle control. Both combinations induced a significant decrease in the cell viability compared to TMZ at all time points (**** = *p* < 0.0001). The cytotoxicity of TMZ/Simva/ASH was significantly higher than ASH at all time points (**** = *p* < 0.0001) (**A**–**D**). Representative of histogram plots of the flow cytometry analysis of apoptosis using propidium iodide (PI) assay (**E**). Numbers indicate the cell percentages in the sub-G1 phase of the cell cycle. All experiments were done in nine replicates and three independent biological replicates. The results were shown as the mean ± SD. We measured apoptosis using a Nicoletti assay at 72 h for U87 and U251 cells. The cell cycle pattern of each treatment is shown for both cell lines. The results showed a significant increase of apoptotic cell populations (sub-G1) in TMZ/Simva- and TMZ/Simva/ASH-treated cells, compared to the TMZ treatment alone in both U87 and U251 cells (**** = *p* < 0.0001). Moreover, apoptosis was reinforced in TMZ/Simva/ASH combination to ASH in GBM cells (** = *p* < 0.01, *** = *p* < 0.001, **** = *p* < 0.0001) (**F**,**G**). Caspase-3/-7 activity was assessed in GBM cells which were treated with TMZ (100 µM), Simva (1µM for U251 and 2.5 µM for U87), and ASH (1.5 µM) at 48 h (**H**,**I**). The activation of caspase-3/-7 was determined by a Cayman fluorescence assay kit. All treatments significantly increased caspase-3/-7 activity to untreated U87 and U251 vehicle time-match controls (**** = *p* < 0.0001). Caspase-3/-7 activation was significantly higher in both combination treatments (TMZ/Simva, TMZ/Simva/ASH) compared to TMZ and ASH treatments, in both U87 and U251 GBM cells (**** = *p* < 0.0001).

**Figure 3 biology-12-00302-f003:**
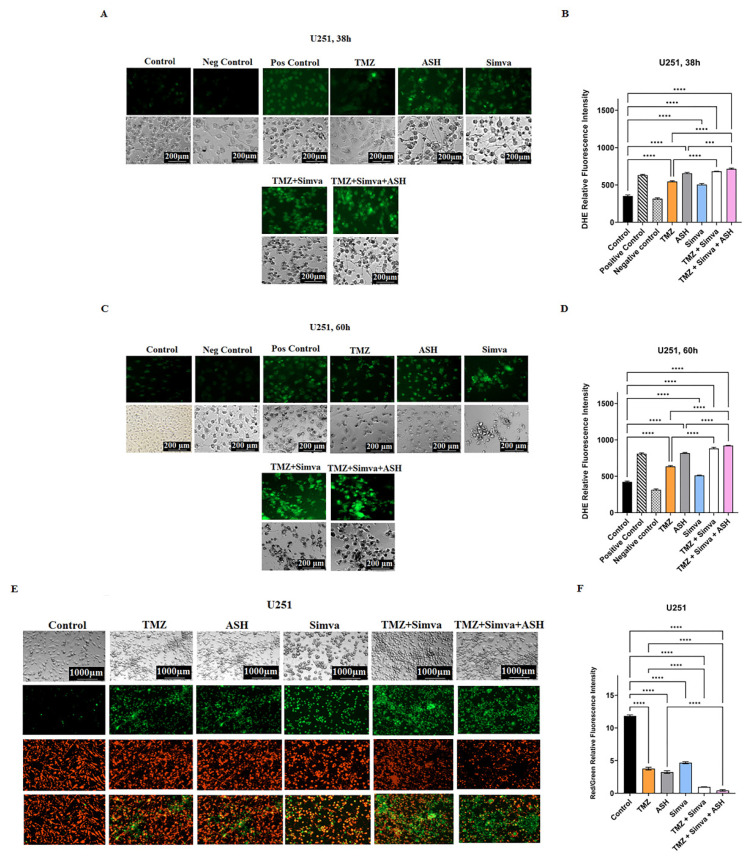
Simvastatin, temozolomide, acetylshikonin and their combination treatments induce mitochondrial damage in human glioblastoma cells. U251 cells were treated with TMZ 100 µM, Simva 1 µM, ASH 1.5 µM, TMZ/Simva and TMZ/Simva/ASH at two different time points (38 and 60 h). A Cayman ROS detection cell-based assay (DHE) was used to assess levels of ROS under different conditions. All treatments induced a significant increase in ROS at both time points compared to the time-match control (**** = *p* < 0.0001). Additionally, both combinations treatments (TMZ/Simva, TMZ/Simva/ASH) significantly induced more ROS compared to TMZ and ASH single treatments at both time points (38 and 60 h) (*** = *p* < 0.001; **** = *p* < 0.0001) (**A**–**D**). We also measured the mitochondrial membrane potential (MMP) in the U251 cell line. All treatments (TMZ 100 µM, Simva 1 µM, ASH 1.5 µM, TMZ/Simva and TMZ/Simva/ASH) induced a significant decrease in MMP compared to the time-match control (**** = *p* < 0.0001) (**E**,**F**). Both combination treatments induced a decrease in MMP compared to TMZ and ASH single treatments (**** = *p* < 0.0001). All experiments have been done in three independent biological replicates.

**Figure 4 biology-12-00302-f004:**
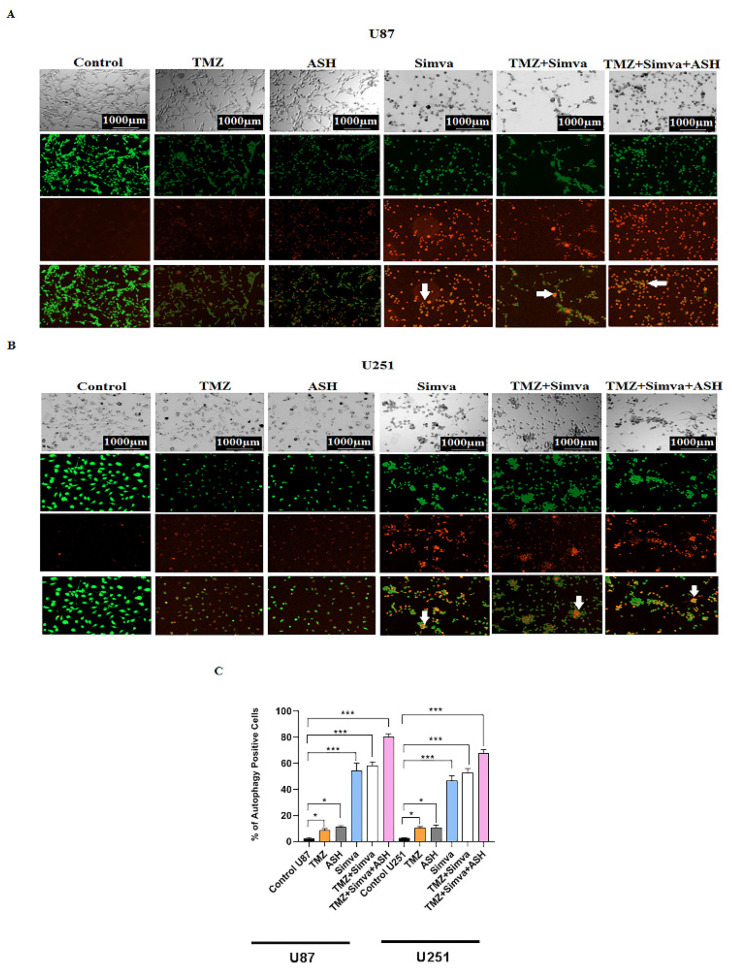
Simvastatin, Temozolomide, and Acetylshikonin combination treatments change autophagy flux in human glioblastoma cells. U87 and U251 cells were treated with TMZ 100 µM, Simva (2.5, 1 µM respectively), ASH 1.5 µM, TMZ/Simva and TMZ/Simva/ASH for 48 h. The formation of acidic vesicular organelles was assessed using acridine orange (AO). The green fluorescence turned to red AVOs in the treatment conditions (white arrows (**A**,**B**)). The percentage of AVO positive cells were evaluated. All treatments significantly increased the percentage of AVO positive cells in both cell lines compared to the time match vehicle control. The cells treated with TMZ/Simva/ASH showed the highest percentage of AVO positive cells (* = *p* < 0.05, *** = *p* < 0.001) (**C**). We further evaluated autophagy in different experimental conditions using immunoblotting. GBM cells were treated with Bafilomycin A1 (autophagy flux inhibitor) (5 nM), ASH, Baf/ASH, TMZ/Simva/ASH and Baf/TMZ/Simva/ASH for 72 h (all concentrations were the same as AO experiments). p62 degradation, LC3β lipidation and formation of LC3β-II and Beclin-1 expressions were evaluated using immune blotting. Ponceau S was utilized as loading control (**D**). ASH increased the turnover of autophagosomes in both U87 and U251 cells, although it was not statistically significant (**E**–**H**,**J**–**N**). Baf significantly increased LC3β-II lipidation and decreased p62 degradation in both U87 and U251 cells, confirming the effect of ASH in autophagy and autophagosome turnover (**E**–**I**;**J**–**M**). TMZ/Simva/ASH partially inhibited autophagy flux, which is confirmed by the non-significant increase of LC3β-II and non-significant decrease in p62 degradation. Baf significantly induced LC3β lipidation, and accumulation of p62 in combination with TMZ/Simva/ASH treatment in both U87 and U251 cells (**E**–**H**,**J**–**M**). Beclin-1 is not involved in ASH, and TMZ/Simva/ASH-induced autophagy in U87 and U251 cells (**I**,**N**). (ns: non-significant; * = *p* < 0.05; ** = *p* < 0.01, *** = *p* < 0.001, **** = *p* < 0.0001).

**Figure 5 biology-12-00302-f005:**
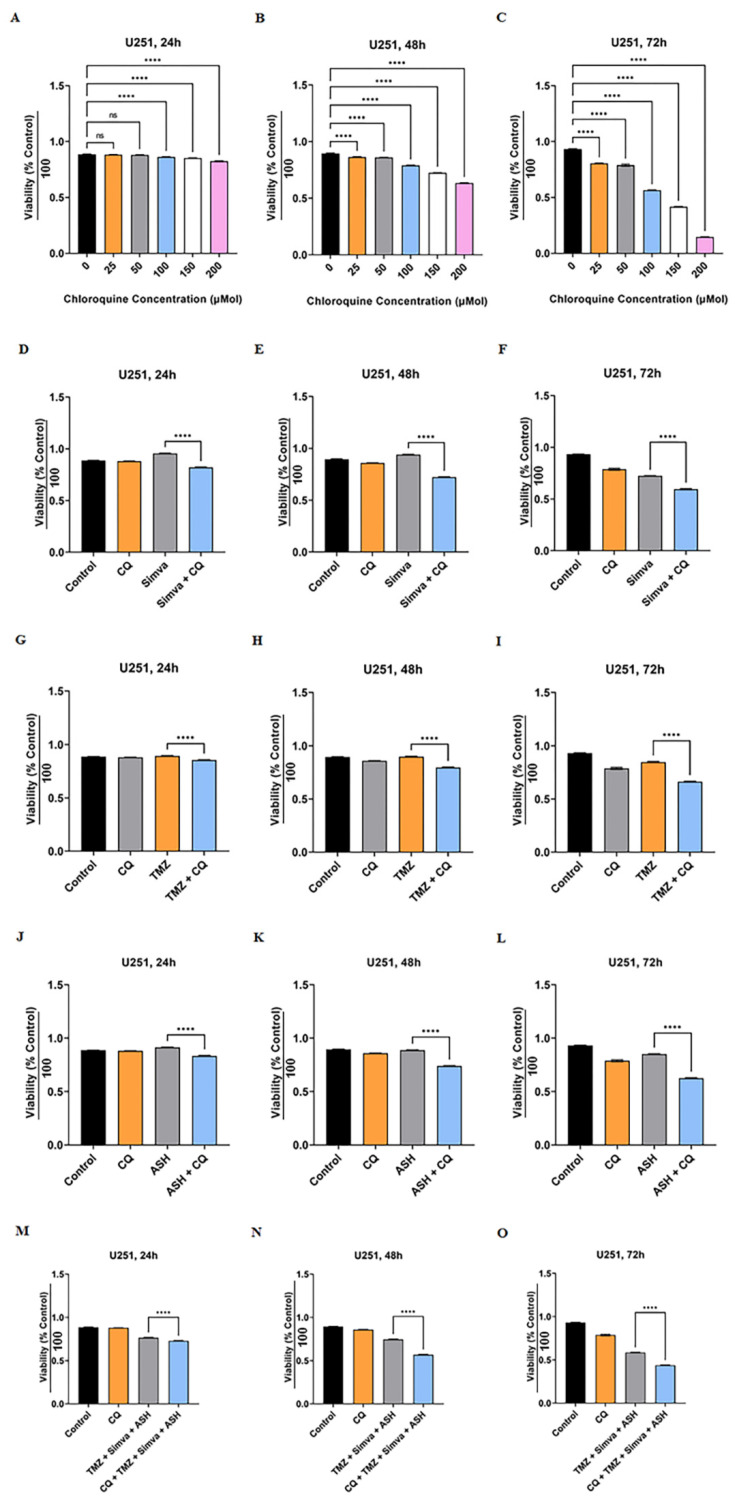
Autophagy flux inhibition increases simvastatin, temozolomide, acetylshikonin and triple-combination treatment-induced cell death. U251 cells were treated with different concentrations of chloroquine (CQ) (25–200 µM) at 24, 48 and 72 h. Cell viability was assessed via an MTT test (**A**–**C**). CQ induced significant cell death for all concentrations at all time points (**** = *p* < 0.0001), except 25 and 50 µM concentrations at 24 h (*p* > 0.05). We used CQ 50 µM in combination with Simva 1 µM (**D**–**F**), TMZ 100 µM (**G**–**I**), ASH 1.5 µM (**J**–**L**) and TMZ/Simva/ASH (**M**–**O**) for 24, 48, and 72 h. CQ decreased cell viability in combination with all treatments (**** = *p* < 0.0001), which showed that TMZ, Simva, ASH, and TMZ/Simva/ASH-induced cell death is dependent on autophagy flux. All experiments have been done in nine replicates and three independent biological replicates, and are showed as the mean ± SD (ns: non-significant).

**Figure 6 biology-12-00302-f006:**
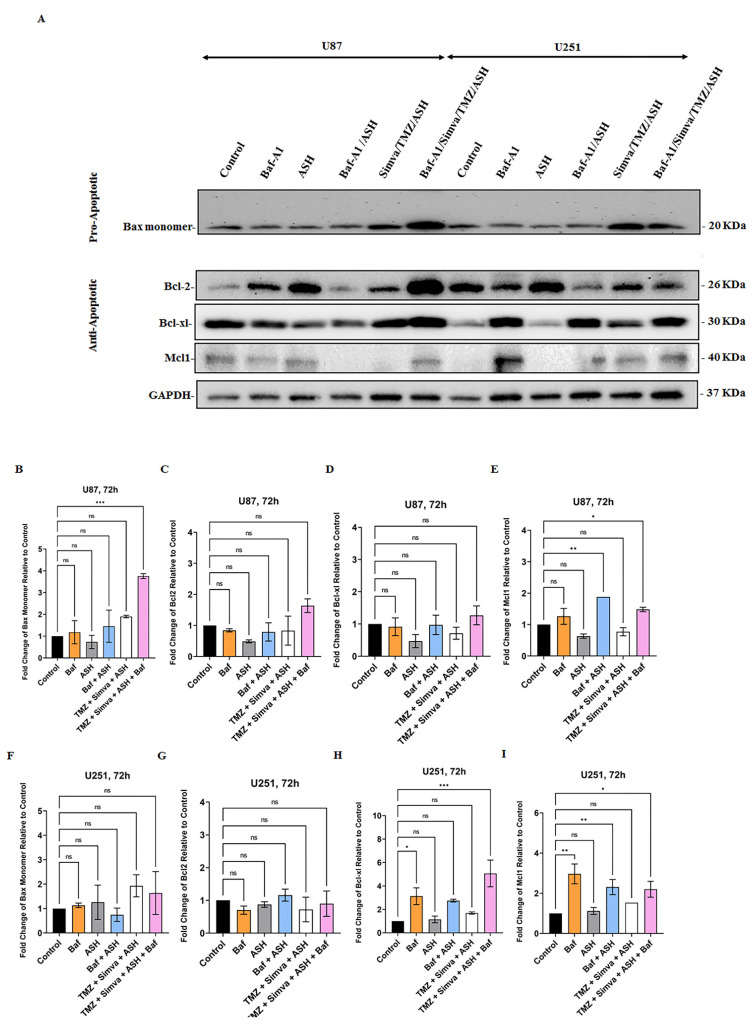
Effect of Autophagy Flux inhibition on pro and anti- apoptotic proteins expressions in GBM cells. We used Bafilomycin A1 (Baf) 5 nM in combination with ASH 1.5 µM and TMZ/Simva/ASH (Simva 1 µM, U251 and Simva 2.5 µM, U87). We evaluated pro- (Bax) and anti- apoptotic Bcl2 (Bcl2, Bcl-XL, Mcl-1) family proteins in different experimental conditions using immunoblotting at 72 h. GAPDH was utilized as loading control (**A**). TMZ/Simva/ASH non-significantly increased Bax expression (*p* > 0.05) while TMZ/Simva/ASH/Baf significantly increased Bax expression in U87 cells (*** = *p* < 0.001), (**B**). All treatments did not significantly change Bcl2 (**C**) and Bcl-XL (**D**) expressions in U87 cells. The Baf treatment significantly increased Mcl-1 expression in both ASH (*p* < 0.01) and TMZ/Simva/ASH (* = *p* < 0.05) treatments in U87 cells (**E**). TMZ/Simva/ASH and its combination with Baf non-significantly increased Bax expression (*p* > 0.05) in U251 cells (**F**). All treatments did not significantly change Bcl2 (**G**) expression in U251 cells. Baf treatment significantly increased Bcl-XL (**H**) expression in U251 cells (single Baf treatment, *p* < 0.05, Baf-TMZ/Simva/ASH, *p* < 0.001). Baf treatment significantly increased Mcl-1 (**I**) expression in U251 cells (single Baf treatment, *p* < 0.01, Baf-TMZ/Simva/ASH, *p* < 0.05). All experiments have been done in 9 replicates and three independent biological replicates and are shown as mean ± SD (ns: non-significant; ** = *p* < 0.01).

## Data Availability

All original data for immune blotting have been uploaded to the MDPI website. The other data presented in this study are available upon request from the corresponding authors.

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
