# Peer review of "Temozolomide, Simvastatin and Acetylshikonin Combination Induces Mitochondrial-Dependent Apoptosis in GBM Cells, Which Is Regulated by Autophagy"

_biology, 2023, doi:10.3390/biology12020302_

Round 1

Reviewer 1 Report

The paper entitled "Temozolomide, Simvastatin and Acetylshikonin Combination Induces Mitochondrial Dependent Apoptosis in GBM Cells which is Regulated by Autophagy" by Hajiahmadi and colleagues explores the potential application of Temozolomide, Simvastatin and Acetylshikonin co-treatment for the therapy of Glioblastoma Multiforme. 

The paper is well-written and structured. The study is relevant, and the methodology employed is adequate for the proposed objectives. Careful reading did not illuminate a significant need for major revisions. The manuscript has a good experimental design and reporting. It supplements the study field with valuable information.

I only encourage the authors to include more information on the molecular mechanisms triggered by Temozolomide and Simvastatin combination (e.g., doi: 10.3892/mmr.2020.11309)

Author Response

The paper entitled "Temozolomide, Simvastatin and Acetylshikonin Combination Induces Mitochondrial Dependent Apoptosis in GBM Cells which is Regulated by Autophagy" by Hajiahmadi and colleagues explores the potential application of Temozolomide, Simvastatin and Acetylshikonin co-treatment for the therapy of Glioblastoma Multiforme.

The paper is well-written and structured. The study is relevant, and the methodology employed is adequate for the proposed objectives. Careful reading did not illuminate a significant need for major revisions. The manuscript has a good experimental design and reporting. It supplements the study field with valuable information.

Answer: We appreciate the positive feedback of the respected reviewer on our article.

I only encourage the authors to include more information on the molecular mechanisms triggered by Temozolomide and Simvastatin combination (e.g., doi: 10.3892/mmr.2020.11309)

Answer: We followed the respected reviewer comments and added statement to identify about mechanism of TMZ/Simva (page 19 and 20, lines: 493-499). We also followed the respected reviewer comments and look at paper was suggested (e.g., doi: 10.3892/mmr.2020.11309) and found out that on that paper TMZ refers to trimetazidine and not Temozolomide and did not match the content of our investigation. But we are thankful of the respected reviewer on the valuable suggestion.

Reviewer 2 Report

Glioblastoma multiforme (GBM) is the most lethal and devastating malignancy with a grim prognosis. Median survival without treatment is 3-5 months and median survival with surgery and combined radiotherapy and chemotherapy with alkylating agents (such as Temozolomide (TMZ)) is 14-16 months. The authors of this study focused on investigation of proven anti-GBM agents, such as TMZ, Simvastatin (Simva) and Acethylshikonin (ASH). Combination of a statin Simva and TMZ for treatment of GBM patients increases their survival. The authors of this study previously demonstrated successful Simva-TMZ action against GBM primary cells derived from patients and GBM cell lines (U87, and U251) through apoptosis induced by inhibition of self-eating mechanism (autophagy) and the unfolded protein response (UPR). Chinese traditional remedy, named Shikonin (SHK), and its derivates, such as ACH,  originate from Lithospermum erythrorhizon roots and exhibit cytotoxic and anti-cancer effects. Recently, dual treatment with SHK and TMZ showed decreasing of GBM growth, migration and glial-to-mesenchymal transition. Current study is an attempt to elucidate the mechanism of the triple combination of TMZ, Simva and ASH against human GBM cells U87, and U251. Cytotoxicity of the single (TMZ, Simva or ASH, alone), double (TMZ and Simva) and triple treatment (TMZ, Simva and ASH) on U87, and U251 was evaluated by MTT assay. Triple combination treatment induced significantly more apoptosis on both GBM cell lines compared to single and dual treatments. Apoptotic nuclei (i.e. sub-G1 cell population) determined by flow cytometry and the activation of Caspase -3 and -7 correlated with higher apoptosis in dual and triple combination treatments for both cell lines. A significant increase of reactive oxygen species (ROS) and decrease of mitochondrial membrane potential in both cell lines upon dual and triple combination treatments. The impact of individual, dual and triple combination treatments on autophagy was estimated by acridine orange (AO) staining of acidic vesicular organelles (AVOs) as markers of late autophagy. The authors showed that triple combination treatment significantly increased the number of AVOs compared to dual and individual treatments in both cell lines. Western Blot analysis revealed that observed increase of AVOs during triple combination treatment was due to partial inhibition of autophagy flux in GBM cell lines (non-significant increase of LC3β-II and non-significant decrease in p62 degradation). Finally, the role of Bcl2 pro-apoptotic (Bax) and anti-apoptotic (Bcl2, Bcl-XL, Mcl-1) proteins in induced cell death in GBM cells upon of individual, dual and triple combination treatments was evaluated by Western Blot analysis. The authors concluded that induced mitochondrial dependent apoptosis in GBM cell lines potentially depends on an increase of Bax protein expression.

The importance of this study is discovery of more effective triple combination therapy of TMZ, Simva and ASH compared to their individual and dual combination therapy impact to GBM cell death. The authors emphasize that this discovery has opened up new fields of research for further identification of the mechanisms involved in novel combination of anti-GBM therapeutic agents. Having in mind the significant deadly impact of novel combination of therapeutic agents on GBM cells and an extensive research methodology involved in this study, I warmly recommend this manuscript for publishing in your journal in present form.

Author Response

Glioblastoma multiforme (GBM) is the most lethal and devastating malignancy with a grim prognosis. Median survival without treatment is 3-5 months and median survival with surgery and combined radiotherapy and chemotherapy with alkylating agents (such as Temozolomide (TMZ)) is 14-16 months. The authors of this study focused on investigation of proven anti-GBM agents, such as TMZ, Simvastatin (Simva) and Acethylshikonin (ASH). Combination of a statin Simva and TMZ for treatment of GBM patients increases their survival. The authors of this study previously demonstrated successful Simva-TMZ action against GBM primary cells derived from patients and GBM cell lines (U87, and U251) through apoptosis induced by inhibition of self-eating mechanism (autophagy) and the unfolded protein response (UPR). Chinese traditional remedy, named Shikonin (SHK), and its derivates, such as ACH,  originate from Lithospermum erythrorhizon roots and exhibit cytotoxic and anti-cancer effects. Recently, dual treatment with SHK and TMZ showed decreasing of GBM growth, migration and glial-to-mesenchymal transition. Current study is an attempt to elucidate the mechanism of the triple combination of TMZ, Simva and ASH against human GBM cells U87, and U251. Cytotoxicity of the single (TMZ, Simva or ASH, alone), double (TMZ and Simva) and triple treatment (TMZ, Simva and ASH) on U87, and U251 was evaluated by MTT assay. Triple combination treatment induced significantly more apoptosis on both GBM cell lines compared to single and dual treatments. Apoptotic nuclei (i.e. sub-G1 cell population) determined by flow cytometry and the activation of Caspase -3 and -7 correlated with higher apoptosis in dual and triple combination treatments for both cell lines. A significant increase of reactive oxygen species (ROS) and decrease of mitochondrial membrane potential in both cell lines upon dual and triple combination treatments. The impact of individual, dual and triple combination treatments on autophagy was estimated by acridine orange (AO) staining of acidic vesicular organelles (AVOs) as markers of late autophagy. The authors showed that triple combination treatment significantly increased the number of AVOs compared to dual and individual treatments in both cell lines. Western Blot analysis revealed that observed increase of AVOs during triple combination treatment was due to partial inhibition of autophagy flux in GBM cell lines (non-significant increase of LC3β-II and non-significant decrease in p62 degradation). Finally, the role of Bcl2 pro-apoptotic (Bax) and anti-apoptotic (Bcl2, Bcl-XL, Mcl-1) proteins in induced cell death in GBM cells upon of individual, dual and triple combination treatments was evaluated by Western Blot analysis. The authors concluded that induced mitochondrial dependent apoptosis in GBM cell lines potentially depends on an increase of Bax protein expression.

The importance of this study is discovery of more effective triple combination therapy of TMZ, Simva and ASH compared to their individual and dual combination therapy impact to GBM cell death. The authors emphasize that this discovery has opened up new fields of research for further identification of the mechanisms involved in novel combination of anti-GBM therapeutic agents. Having in mind the significant deadly impact of novel combination of therapeutic agents on GBM cells and an extensive research methodology involved in this study, I warmly recommend this manuscript for publishing in your journal in present form.

Answer: We appreciate the positive feedback and trust of the respected reviewer to our article.

Reviewer 3 Report

This manuscript evaluates in vitro the efficacy of a tri-therapy that includes Temozolomide, Simvastatin, and Acetylshikonin, promoting the decrease of cancer cell viability through different mechanisms, including apoptosis and autophagy. The authors used two cancer cell lines to reach their goals, namely U87 and U251. They used multiple complementary techniques to demonstrate that this combined drug could kill cancer cells at a better level than if used individually. Two key pathways were involved in such effects on cancer cells. These pathways involve apoptosis through caspase 3/7 activation and autophagy through Bax.

This study is well-designed and could interest the readers of this journal.

Minor comments:

1. The authors could comment on the specificity of these three molecules affecting cancer cells but not in normal cells.

2. What would be the efficient time for these molecules to control/irradicate the cancer cells? A short or a long-time treatment?

3. Can we anticipate certain adverse effects using this tri-therapy?

4. This team listed too many references from their work. For example, in section 2.3 Cytotoxicity assay, there were 6 references listed to show that they have the MTT assay optimized in their labs. It is unnecessary to record such a high number of already published papers about such a technique. Similar remarks apply all over the manuscript.

5. There are too many references in the results section. These refs, even if they are from the same research group, should be moved to the discussion. The results section should be focussed on what has been found and is being presented in the manuscript.

6. Fig. 1, Y axis should be in % as the axis identification is (viability (% Control).

7. Similar remark for Fig. 2 (A, B, C, D)

Author Response

This manuscript evaluates in vitro the efficacy of a tri-therapy that includes Temozolomide, Simvastatin, and Acetylshikonin, promoting the decrease of cancer cell viability through different mechanisms, including apoptosis and autophagy. The authors used two cancer cell lines to reach their goals, namely U87 and U251. They used multiple complementary techniques to demonstrate that this combined drug could kill cancer cells at a better level than if used individually. Two key pathways were involved in such effects on cancer cells. These pathways involve apoptosis through caspase 3/7 activation and autophagy through Bax.

This study is well-designed and could interest the readers of this journal.

Answer: We appreciate the positive feedback and trust of the respected reviewer to our article.

Minor comments:

  1. The authors could comment on the specificity of these three molecules affecting cancer cells but not in normal cells.

Answer: We are thankful for great question to increase more validity to our article. We added a paragraph (page 21, lines: 563-566 to address the respected reviewer question. As all of these compounds have effects on normal cells, we will use nano-delivery of these compounds for decreasing side effects of these compounds.

  1. What would be the efficient time for these molecules to control/irradicate the cancer cells? A short or a long-time treatment?

Answer: We appreciate the important question of the respected reviewer. Actually TMZ/SImva/ASH combination treatment induces faster apoptosis and cell death compared to TMZ based on the difference in mechanism. We added this point to the conclusion of the revised manuscript (page 21, line 580-581).

  1. Can we anticipate certain adverse effects using this tri-therapy?

Answer: We appreciate the important question of the respected reviewer. Actually, this is not something that we can answer now and needs further investigations.  

  1. This team listed too many references from their work. For example, in section 2.3 Cytotoxicity assay, there were 6 references listed to show that they have the MTT assay optimized in their labs. It is unnecessary to record such a high number of already published papers about such a technique. Similar remarks apply all over the manuscript.

Answer: We appreciate the the important questions. The reason we have included our established references in the methods it that we want to give the confidence to respected authors about the reproducibility of all results. But we followed the respected reviewer suggestions and excluded some of the references from the results.

  1. There are too many references in the results section. These refs, even if they are from the same research group, should be moved to the discussion. The results section should be focussed on what has been found and is being presented in the manuscript.

Answer: We appreciate the respected reviewer feedback. We tried to give the rational for each experiments that has been done and the results are provided. This is way we wrote an introductory section in the beginning of each results and include some related previous work and then explained the results. It gives more understanding of the rational of the experiments to the readers.

  1. Fig. 1, Y axis should be in % as the axis identification is (viability (% Control).

Answer: We appreciate the respected reviewer question. Y axis is Viability (% control) so the number is shown in the Y axis is %. Therefore the current format is correct.

  1. Similar remark for Fig. 2 (A, B, C, D)

Answer: We appreciate the respected reviewer question. Y axis is Viability (% control) so the number is shown in the Y axis is %. Therefore the current format is correct.